# Effect of the Size and Shape of Ho, Tm:KLu(WO_4_)_2_ Nanoparticles on Their Self-Assessed Photothermal Properties

**DOI:** 10.3390/nano11020485

**Published:** 2021-02-14

**Authors:** Albenc Nexha, Maria Cinta Pujol, Joan Josep Carvajal, Francesc Díaz, Magdalena Aguiló

**Affiliations:** Campus Sescelades, Física i Cristalografia de Materials i Nanomaterials (FiCMA-FiCNA)-EMaS, Departament de Química Física i Inorgànica, Universitat Rovira I Virgili, E-43007 Tarragona, Spain; albenc.nexha@urv.cat (A.N.); mariacinta.pujol@urv.cat (M.C.P.); f.diaz@urv.cat (F.D.); magdalena.aguilo@urv.cat (M.A.)

**Keywords:** solvothermal synthesis, monoclinic double tungstates, nanocrystals, luminescence nanothermometry, photothermal conversion efficiency, third biological window spectral regime

## Abstract

The incorporation of oleic acid and oleylamine, acting as organic surfactant coatings for a novel solvothermal synthesis procedure, resulted in the formation of monoclinic KLu(WO_4_)_2_ nanocrystals. The formation of this crystalline phase was confirmed structurally from X-ray powder diffraction patterns and Raman vibrational modes, and thermally by differential thermal analysis. The transmission electron microscopy images confirm the nanodimensional size (~12 nm and ~16 nm for microwave-assisted and conventional autoclave solvothermal synthesis) of the particles and no agglomeration, contrary to the traditional modified sol-gel Pechini methodology. Upon doping with holmium (III) and thulium (III) lanthanide ions, these nanocrystals can generate simultaneously photoluminescence and heat, acting as nanothermometers and as photothermal agents in the third biological window, i.e., self-assessed photothermal agents, upon excitation with 808 nm near infrared, lying in the first biological window. The emissions of these nanocrystals, regardless of the solvothermal synthetic methodology applied to synthesize them, are located at 1.45 μm, 1.8 μm and 1.96 μm, attributed to the ^3^H_4_ → ^3^F_4_ and ^3^F_4_ → ^3^H_6_ electronic transition of Tm^3+^ and ^5^I_7_ → ^5^I_8_ electronic transition of Ho^3+^, respectively. The self-assessing properties of these nanocrystals are studied as a function of their size and shape and compared to the ones prepared by the modified sol-gel Pechini methodology, revealing that the small nanocrystals obtained by the hydrothermal methods have the ability to generate heat more efficiently, but their capacity to sense temperature is not as good as that of the nanoparticles prepared by the modified sol-gel Pechnini method, revealing that the synthesis method influences the performance of these self-assessed photothermal agents. The self-assessing ability of these nanocrystals in the third biological window is proven via an ex-vivo experiment, achieving thermal knowledge and heat generation at a maximum penetration depth of 2 mm.

## 1. Introduction

Photothermal therapy, that employs light-absorbing agents to convert photoenergy into heat, achieving local hyperthermia, is regarded as a minimally invasive and highly efficient methodology for targeted cancer treatment [1,2,3,4]. Efficient photothermal agents require high light-to-heat conversion efficiency, small size, and working under excitation and generated emission wavelengths operating in the optical biological windows [4]. To these, it can be added real-time temperature feedback to monitor the treatment [4].

A high light-to-heat conversion efficiency allows performing effective hyperthermia with low laser powers and suppresses any overheating effect [4]. Small sizes of the photothermal agents guarantee successful intracellular studies and applications, and the effective excretion of the nanoparticles after the photothermal treatment, avoiding in-vivo toxicity [1,2,3,4].

Regarding the excitation wavelength used, it should exhibit deep penetration depths within the biological tissues to trigger these agents inside the human body by exciting them from outside or through laparoscopic approaches [4]. Energy dissipation in biological tissues arise from light absorption and scattering by biological tissue components (mainly water, fat, hemoglobin, and melanin) [5]. Thus, this energy dissipation should be minimized. High penetration depths are achieved when the used light (both for excitation and from generated emissions) lies in the so-called optical biological windows. Biological windows are the spectral ranges where biological tissues become partially transparent due to a simultaneous reduction in both absorption and scattering of light [5]. They are split into four different spectral ranges, apart one of the other mainly by the different absorption bands of water: (i) the first biological window (I-BW) extending from 650 nm to 950 nm; (ii) the second biological window (II-BW) extending from 1000 nm to 1350 nm; (iii) the third biological window (III-BW) extending from 1350 nm to 2000 nm; and (iv) the fourth biological window (IV-BW), centered at 2200 nm [6]. Among these biological windows, the III-BW achieves the deepest light penetration in specific biological tissues like those containing melanin and oxygenated blood, due to an effective light transmission [6], and a maximum reduction of the scattering coefficient [7].

Laser-triggered photothermal therapy using near infrared excitation sources (NIR, λ = 700–1100 nm) is highly attractive due to deeper penetration in biological tissues, compared to ultraviolet (UV) or visible (VIS) light sources [8,9]. Photothermal agents that strongly respond to light excitation with NIR wavelengths and generate emissions within the biological windows can lead to non-invasive real-time temperature sensing in situ [1,2,3,4]. 

Materials including gold nanostructures (nanostars [10], nanorods [10,11], nanoshells [12] and nanocages [13]), carbon [14,15,16,17], palladium [18,19,20,21] and copper [22,23,24] based nanomaterials, have been reported as potential photothermal agents. Despite their considerable high light-to-heat conversion efficiency, particularly for the gold nanostructures, these photothermal agents require additional components able to measure temperature due to the lack of self-assessment feedback [25]. This feedback is especially important when the real temperature inside the tumor is substantially different from the one that can be determined from outside the body [26]. Hence, multifunctional agents that combine photothermal conversion efficiency and thermal sensing, are highly desirable.

Lanthanide (Ln^3+^)-doped nanomaterials offer both these properties in a single material due to their peculiar electronic configuration, giving rise to radiative and non-radiative processes, responsible for the generation of luminescence and heat, respectively [27,28,29]. These two properties are exhibited simultaneously in a material doped with Ln^3+^ ions, after excitation with the suitable wavelength, rendering this material the potential to be self-assessed photothermal agent, i.e., the same material releases heat and emits light that allows determining the temperature in situ. Holmium (Ho^3+^) and thulium (Tm^3+^) doped potassium lutetium double tungstates KLu(WO_4_)_2_ (hereafter Ho, Tm:KLuW) after being illuminated with NIR light exhibit this self-assessed photothermal property [29]. Previous results have confirmed that this material exhibit potential thermal sensing properties in the III-BW and light-to-heat conversion efficiency ~40% [29]. The multifunctionality of Ho, Tm:KLuW nanoparticles was demonstrated by determining the temperature in situ in an ex-vivo experiment increasing the temperature of the biological sample by 17 K [29]. However, the dual functionality of Ho, Tm:KLuW as thermal sensing and photothermal conversion agents was demonstrated on nanocrystals synthesized by the modified sol-gel Pechini method. This methodology often suffers from morphological irregularities and wide size distribution of the particles up to several microns [28,29,30]. Hence, engineering and developing this material to control its morphology and size dispersion to generate potential self-assessed photothermal conversion agents, is important. A possibility to do this is by synthesizing non-aggregated, well-shaped and nanodimensional Ho, Tm:KLuW particles by applying novel microwave-assisted and conventional autoclave solvothermal methodologies in the presence of organic surfactants, such as oleic acid (OLAC) and oleylamine (OLAM), to control their nucleation and growth.

The incorporation of OLAC and OLAM acting as surfactants, either alone, combined or mixed with 1-octadecene (ODE) or trioctylphosphine oxide (TOPO), has allowed a good control on the synthesis of the inorganic colloidal nanocrystals (oxides, sulfides, tellurides, chalcogenides (binary, ternary and quaternary), and metallic nanocrystals [31], via wet chemical methodologies. The combinatory effect of OLAC-OLAM has also been used in the synthesis of Ln^3+^-doped materials (sesquioxides [32], tetragonal double tungstates [33,34] and fluorides [35]). However, the major part of the methodologies with this binary combination are applied in heat up (or thermal decomposition) methodologies, whereas reports in the application of this combination in microwave-assisted or solvothermal synthesis methodologies are scarce, especially when dealing with Ln^3+^ doped materials. To the best of our knowledge, the only report so far for the implementation of the combinatory effect of OLAC and OLAM surfactants in microwave-assisted or hydrothermal/solvothermal methodologies for the synthesis of Ln^3+^ doped material, is attributed to the preparation of tetragonal bipyramids of Eu^3+^:NaLa(MoO_4_)_2_ [36]. An equimolar ratio of OLAC:OLAM was incorporated into this solvothermal approach at 413 K during 6 h.

Here, undoped KLuW and Ho-, Tm-doped KLuW nanocrystals with sizes below 20 nm have been synthesized via novel solvothermal methodologies (microwave-assisted and conventional autoclave). The successful formation of the monoclinic crystalline phase was confirmed by X-ray powder diffraction (XRPD), Raman vibrational modes and differential thermal analysis (DTA). A possible mechanism underlying the combined effect of OLAC and OLAM surfactants is proposed. The effect of the nanodimensional size and ellipsoidal shape of Ho, Tm;KLuW nanocrystals synthesized by these methodologies on their self-assessed photothermal properties with thermal sensing in the III-BW by luminescence thermometry, is investigated. The emissions of these nanocrystals lying in the III-BW are generated after excitation with 808 nm NIR light, lying in the I-BW. For thermal sensing, the intensity ratio between the 1.45 μm and 1.8 μm emission bands of Tm^3+^ and the 1.96 μm emission band of Ho^3+^ is analyzed in the physiological range of temperatures (293–333 K). The photothermal conversion efficiency, determined by the integrated sphere method, [37] was also investigated using the 808 nm energy source. The multifunctional properties of these nanocrystals, were demonstrated in an ex-vivo experiment, confirming the ability of these materials to simultaneously sense temperature and generate heat through NIR excitation.

## 2. Materials and Methods 

### 2.1. Materials

Lutetium nitrate hydrate (Lu(NO_3_)_3_·H_2_O, 99.99%), holmium nitrate pentahydrate (Ho(NO_3_)_3_·5H_2_O, 99.9%), potassium tungstate (K_2_WO_4_, 99.5%), oleylamine (OLAM, 70%), potassium carbonate (K_2_CO_3_,99.99%) and n-hexane (99%) were purchased from Alfa Aesar (Haverhill, MA, US). Thulium nitrate pentahydrate (Tm(NO_3_)_3_·5H_2_O, 99.9%) and oleic acid (OLAC, 90%) was purchased from Sigma Aldrich (San Luis, MO, US). Absolute ethanol was purchased from VWR Chemicals (Radnor, PA, US).

### 2.2. Synthesis of Nanocrystals

#### 2.2.1. Microwave-Assisted Solvothermal Synthesis of Undoped and Doped KLu(WO_4_)_2_ Nanocrystals

The microwave-assisted solvothermal reaction was carried out in an ETHOS One reactor (Milestone Srl, Sorisole, Italy) equipped with temperature sensor and a stirring bar. The stoichiometric ratio of the precursors and the ratio of the precursors versus the organic surfactants, have been chosen taking as reference the work of Bu et al. [36] in which they prepared nanocrystals of double molybdates NaLa(MoO_4_)_2_. The initial stoichiometric ratio of the precursors and the ratio of the precursors versus the organic surfactants, were inspired in the above mentioned work. In the present work, we studied the effects on the products of the reduction of the reaction temperature and time in the microwave-assisted solvothermal approach, with the aim to develop a green approach for the synthesis of KLu(WO_4_)_2_. Other parameters, such as the different cation precursors used, the OLAC:OLAM molar ratio, and the calcination temperature and time were also explored, as they are summarized in Appendix A.

In a typical microwave-assisted solvothermal synthesis (hereafter MW) of the undoped KLuW nanoparticles with the optima conditions, OLAC (6 × 10^−3^ mol) and OLAM (6 × 10^−3^ mol) were mixed in 18 mL of ethanol and were magnetically stirred for 20 min until a homogeneous transparent solution was obtained. The starting OLAC:OLAM molar ratio was 1:1, as used in the work of Bu et al. [36]. Two different solutions in 4 mL of water were prepared containing 0.3 × 10^−3^ mol of Lu(NO_3_)_3_·H_2_O and 0.6 × 10^−3^ mol of K_2_WO_4_, respectively. K_2_WO_4_ was used as the K^+^ cation and WO_4_^2−^ anion precursor, simultaneously, implying an initial stoichiometry between the cations of K:Lu:W = 4:1:2. The aqueous solution of Lu(NO_3_)_3_·H_2_O was added slowly to the organic-based surfactant solution while stirring, which turned from a transparent homogeneous solution to a milky one. When the K_2_WO_4_ precursor aqueous solution was added dropwise, the milky mixture’s color was slightly attenuated, mainly by the addition of water. All the mixture was transferred to a microwave vial with a total volume of 70 mL, equipped with a stirring bar, which was filled up to 38% of its total volume. The reaction was performed at 453 K for 3 h with stirring. This reaction temperature was reached in 10 min. After the reaction, the system was cooled down to room temperature naturally. The product of this reaction (hereafter the seeds) were collected after purifying the mixture with an excess of ethanol to extract the precipitate, centrifuged at 6000 rpm for 5 min, and redispersed in *n*-hexane. This purification step was repeated until the supernatant was colorless and transparent, removing in this way the excess of organic surfactants. The seeds were obtained in powder form after treatment at 353 K for 1 h, approximately. The seeds were calcined in a conventional furnace at 1023 K for 2 h to obtain KLuW as the final product. For the synthesis of Ho, Tm doped KLuW, the methodology was the same, except the addition of 3 mol% Ho^3+^, 5 mol% Tm^3+^ or 1 mol% Ho^3+^, 10 mol% Tm^3+^, as dopants substituting the Lu^3+^ ions in the KLuW host. These doping ratios ratio represent the maximum brightness of both emissions in the III-BW and the best self-assessing properties, as presented in our previous work [29]. 

#### 2.2.2. Conventional Autoclave Solvothermal Synthesis of Undoped and Doped KLu(WO_4_)_2_ Nanocrystals

The conventional autoclave solvothermal synthesis of undoped and doped KLuW was performed using the same experimental parameters as in the microwave-assisted synthesis, but changing the solution container (here it is used an autoclave metallic reactor) and the heating was provided by a conventional heating furnace. In this methodology, the reaction time and temperature were also optimized, as summarized in Appendix A.

In a typical conventional autoclave solvothermal synthesis (hereafter CA) of the undoped KLuW with the optima conditions, the autoclave metallic reactor, which has a total volume of 60 mL, was filled up to 38% of the total volume, in order to keep the same conditions as in the microwave-assisted reaction. The purification process applied to the product obtained in the reaction of conventional heating solvothermal synthesis was the same than that used in the microwave assisted synthesis. The calcination process was set at 1023 K and 2 h.

### 2.3. Characterization

X-ray powder diffraction (XRPD) measurements were made using a D5000 diffractometer (Siemens, Munich, Germany), using Bragg-Brentano parafocusing geometry and a vertical θ-θ goniometer) fitted with a curved graphite diffracted-beam monochromator, incident and diffracted-beam Soller slits, a 0.06° receiving slit and scintillation counter as detector. The angular 2θ diffraction range was set between 5 and 70°. The data were collected with an angular step of 0.05° at 3 s per step. Cu K_α_ radiation was obtained from a copper X-ray tube operated at 40 kV and 30 mA. The temperature-dependent X-ray powder diffraction measurements were made using an AXS D8-Discover diffractometer (Bruker, Billerica, MA, US) equipped with parallel incident beam (Göbel mirror), vertical θ-θ goniometer, XYZ motorized stage and a General Area Diffraction System (GADDS) as detector. Samples were placed directly on the sample holder and the area of interest was selected with the aid of a video-laser focusing system. An X-ray collimator system allows to analyse areas of 500 μm. The X-ray diffractometer was operated at 40 kV and 40 mA to generate Cu K_α_ radiation. The GADDS detector was a HI-STAR (multiwire proportional counter of 30 × 30 cm^2^ with a 1024 × 1024 pixel). We collected one frame (2D XRD patterns) covering 2θ = 12–45° with a detector-sample distance of 15 cm. The exposition time was 300 s per frame and it was γ-integrated to generate the conventional 2θ vs. intensity diffractogram. Identification of the crystal phases was achieved by comparison of the XRD diffractogram with the ICDD data base (release 2007) using Diffrac^plus^ Evaluation software (Bruker 2007). The temperature was controlled with a MRI BTS-Solid temperature sample stage (Pt heating ribbon heating stage). The patterns were collected from room temperature up to 1273 K, heating at a rate of 10 K/min. The sample was maintained during 60 s at the desired temperature before starting the measurement. The temperature stage was covered with a beryllium (Be) dome and an air static atmosphere was used throughout the measurement.

For the morphological characterization, transmission electron microscopy (TEM) images were acquired using a JEM-1011 electron microscope (JEOL Ltd, Tokyo, Japan) operating at an accelerating voltage of 100 kV. For the preparation of the TEM grids, the seeds dispersed in n-hexane and the final products in distilled water, were placed on the surface of a copper grid covered by a holey carbon film (HD200 Copper Formvar/carbon). 

Fourier Transform Infrared (FT-IR) spectra were recorded in the range of 400–4000 cm^−1^ on a FT-IR IluminatIR II spectrophotometer (Smiths Detection, London, UK) to investigate the presence of the different functional groups on the surface of the samples. 

Differential thermal analysis (DTA) and thermal gravimetric analysis (TGA) were used to study the thermal evolution of the seeds and final products with the temperature by using a SDT 2960 simultaneous differential scanning calorimetry-thermogravimetric analysis system (TA Instruments, New Castle, DE, US). The heating rate was set at 10 K min^−1^ with an air flux of 90 cm^3^ min^−1^. Al_2_O_3_ was used as the reference material.

To characterize the vibration modes of the nanocrystals, a micro-Raman analysis was performed, using an inVia Reflex microscope (Renishaw Plc, Gloucestershire, UK) equipped with unpolarized light coming from a 514 nm argon laser, and focused through a 50× objective (Leica Camera AG, Wetzlar, Germany). The analysis were performed in the range of 200–2000 cm^−1^, using a grating with 2400I mm^−1^ to disperse the spectra and an exposure time of 10 s.

For the photoluminescence analysis of the nanocrystals, the emission spectra were recorded in an AQ6375 optical spectrum analyzer (Yokogawa, Tokyo, Japan) in the range from 1350 nm to 2200 nm, with a resolution of 0.5 nm, a level of accuracy of 1 dB, and an integration time of 1 s. The nanoparticles were excited by the 808 nm emission fiber-coupled diode laser with a power of 200 mW and the beam was focused on the sample using a 20× microscope objective (numerical aperture 0.4) and a spot size of ~10^−6^ m. The excitation density was around 100 W cm^−2^. The scattered excitation radiation was eliminated by using a 850 nm longpass dichroic filter (Thorlabs, Newton, NJ, US). For analyzing the temperature-photoluminescence dependence, the methodology was the same, except that the nanocrystals were located inside a heating stage (THMS 600, Linkam, Tadworth, UK) equipped with a boron disk for improved temperature distribution.

The photothermal conversion efficiency was investigated by applying the method of the integrating sphere [37]. A 4P-GPS-010-SL integrating sphere (Labsphere Inc, North Sutton, NH, US) with an inner diameter of 1 inch, and 4 ports located at 90° one to each other with a diameter of 0.25 inches was used for this purpose. A glass cuvette containing an aqueous solution of the Ho, Tm doped KLuW nanoparticles with a concentration of 1 g L^−1^ was placed on the input port of the integrating sphere, in a position perpendicular to the laser irradiation provide by the 808 nm fiber-coupled diode laser with a power of 200 mW. The laser from the fiber tip was collimated to a spot size of 5 mm in diameter on the sample. A baffle was introduced in the integrating sphere, between the sample and the detector, in order to prevent the direct reflections from the sample to the detector. The generated signals, such as the scattered, reflected, transmitted light and part of the light that is absorbed and converted into another light wavelength, were collected by an Ophir Nova II powermeter [37].

For the ex-vivo experiment, the methodology applied was exactly the same as in the photoluminescence analysis of the nanocrystals, with the addition of a digital multimeter equipped with a platinum and platinum-10% rhodium thermocouple to monitor the temperature, covered with a 2 mm thick chicken breast slice, and placed close to the injected nanoparticles. A thermal camera was also coupled to this optical system to monitor the temperature increase at the surface of the chicken breast.

## 3. Results and Discussion

### 3.1. Characterizations of the Nanocrystals

The synthesis of the undoped KLuW and Ho, Tm doped KLuW nanocrystals, was achieved via microwave-assisted (MW) and conventional autoclave (CA) solvothermal methodologies in the presence of an equimolar ratio of OLAC and OLAM, and the precursors Lu(NO_3_)_3_·H_2_O and K_2_WO_4_, at 453 K for 3 h. This resulted in the formation of the first initial product, that we called seeds, followed by a calcination post treatment at 1023 K during 2 h to form the desired compound. The synthesis conditions were analyzed to optimize the different parameters. The results of this optimization process can be found in the Appendix A. X-ray powder diffraction (XRPD) data reveal that the synthesized nanocrystals crystallize in the monoclinic system with the C2/c spatial group, as confirmed by the comparison with the reference pattern of KLu(WO_4_)_2_ (JCPDS file 54-1204) [38], presented in Figure 1a for the undoped nanocrystals and in Appendix A for the doped nanocrystals. The presence of the dopants displays no significant effect on the purity of the crystalline phase obtained. 

To understand in more detail the physical process that take place during the calcination step, transforming the seeds into the monoclinic KLuW crystalline phase, a Differential Thermal Analysis (DTA) and Thermal Gravimetric Analysis (TGA) of the seeds obtained by the MW solvothermal method, were performed. The thermograms of the seeds are provided in Figure 1b. The seeds were heated from room temperature to 1273 K. During the heating process from room temperature to 800 K, two peaks in the DTA curve were observed, associated to two weight losses in the TGA curve.

The broad peak at 543 K and the less intense peak at 673 K are most probably attributed to the degradation of the organic surfactants [40], which have the ability to bind to the surface of the seeds to prevent their aggregation. In this range of temperatures, the major loss of weight (~4%) is observed, as can be seen in the TGA signal. Similar results are reported for nanoparticles coated with oleic acid, in which these two distinct peaks observed in the DTA curve are related to either the number of layers (bilayer or quasi-two-layers) of oleic acid coating the surface of the nanocrystals or the type of bonding that oleic acid forms with the metal ions [41,42,43]. Instead, if the organic surfactant would be coating the nanoparticles with a monolayer, a single weight loss should be observed, as previously described in the literature [44]. Hence, these results confirm that the seeds are coated with a bilayer or quasi-two-layers of oleic acid. A more complex peak structure appears in the region of 823–933 K, which can be attributed to different transformations of the seeds. The formation of the monoclinic KLuW phase is observed by the appearance of a sharp peak at 1013 K.

To confirm the hypothesis expressed in the previous paragraph we analyzed the evolution of the diffraction pattern of the seeds as a function of temperature in the range from RT to 1273 K. From the XRPD data (Figure 1c) obtained when applying a thermal treatment to the seeds obtained by the MW solvothermal reaction, we can observe that the XRPD patterns recorded from room temperature to 673 K (see also Appendix A), exhibit no significant changes. Instead of analyzing the whole 2θ range, we focused to the 2θ range from 22° to 39°, since it represents the region in which the most intense peaks of both, seeds and final products appear. The XRPD pattern obtained at 973 K, show peaks that can be attributed to the starting seeds and peaks of KLuW, which gradually increased their intensity as the temperature increased. Monoclinic KLuW seems to form at 1013 K, as confirmed from the DTA exothermic peak at this temperature, and it remains as the only crystalline phase present in the diffraction patterns until at least 1173 K, as presented from Figure 1c. At 1273 K, the orthorhombic phase of KLuW is observed, corresponding to the high temperature crystalline phase of this compound [45,46]. This phase would start to form at 1218 K, according to the DTA analysis (Figure 1b).

As a final confirmation for the successful formation of monoclinic KLuW phase, we performed a DTA analysis of the nanoparticles obtained by the MW solvothermal synthesis. The obtained data are compared with the DTA analysis of a bulk monoclinic KLuW crystal, presented in Figure 1d. The two samples have a similar behavior, showing an intense endothermic peak at around 1311–1316 K. As Kletsov et al. [47] indicated the polymorphous transformation in double tungstates takes place at temperatures near the melting point. Hence the transformation from the low temperature crystalline phase (monoclinic) to a higher temperature crystalline phase (orthorhombic) and the melting process are responsible for the appearance of these peaks around 1311–1316 K. These processes are reversible, because KLuW melts congruently, and the crystalline transformation is also reversible [47,48]. The slight difference between the two samples lies on that the peaks of the sample prepared by the MW solvothermal method appear to be slightly shifted to lower temperatures. For example, the endothermic sharp peak observed for the sample coming from the bulk single crystal appears at 1316 K, whereas for the MW solvothermal sample, it appears at 1311 K. Also, during the cooling process the peak observed at 1293 K for the bulk sample, appears at 1273 K for the MW solvothermal sample. These differences may be attributed to the size of the particles in each case (microsize for the bulk crystal sample and nanosize for the MW solvothermal sample). 

Unpolarized Raman spectra were recorded to investigate the vibrational modes of doped KLuW materials. Regardless of the doping ratio used, the observed vibrational modes for all materials, are the expected ones for the monoclinic KLuW phase (Figure 2) [48]. In general, the most intense peak is observed at around 900 cm^−1^ which is attributed to the stretching mode of (W-O), followed by the second most intense peak at around 750 cm^−1^ is the coupling between the stretching mode of (W-O) and oxygen-doubled bridges (WOOW) [48]. The range between 270–400 cm^−1^ is attributed to the WOOW and W-O bending modes [48], and the 400–1000 cm^−1^ range is related to the stretching modes of the WO_6_ group in the double tungstates [48]. The phonons below 270 cm^−1^ are associated to the translational modes of the cations (K^+1^, Lu^3+^ and W^6+^) and the rotational motion of the WO_6_ groups in the unit cell [48]. 

The size and shape of these nanocrystals was investigated by transmission electron microscopy (TEM). The synthesized particles exhibit an ellipsoidal (Figure 3a) and an irregular shape (Figure 3b) for the MW and CA based methodologies, respectively. In terms of the size, the nanocrystals synthesized by MW methodology are smaller compared to the ones produced by the CA methodology, with an average diameter of 12 nm (Figure 3c) for the particles produced by the MW method and a length of 16 nm (Figure 3d) for the particles synthesized by the CA method, respectively. These sizes are similar to the ones determined by applying the Debye-Scherrer equation [49], (19 nm and 28 nm for MW and CA, respectively, Appendix A).

Overall, these solvothermal methodologies not only surpass the limitation exhibited by the traditional modified sol-gel Pechini method regarding the agglomeration and wide size distribution [28,29,30], but in addition offer time and power-effective methods for the synthesis of monoclinic KLuW. In terms of time, the Pechini approach requires approximately one day and a half to prepare the KLuW particles (one day to have the lanthanide precursors completely dissolved in an aqueous solution of EDTA, K_2_CO_3_, and (NH_4_)_2_WO_4_, 2–3 h to evaporate the water after the addition of polyethylene glycol (PEG), followed by 3 h of precalcination process, and finally 2 h of calcination at 1023 K) [28,29,30]. For the solvothermal methodologies, the maximum time needed to prepare the nanocrystals is approximately 6 h, considering here a reaction time of 3 h, the purification process of 20 min, the complete evaporation of the organic apolar solvent (n-hexane) of around 1 h, and the final step of calcination of 1.5–2 h. This time can be reduced even at only 3 h, by considering a reaction in which the sacrificial seeds are obtained in a reaction time of only 0.08 h. However, the final KLuW product obtained for a reaction time of 0.08 h suffers from agglomeration and relatively large particle sizes (Appendix A), not fulfilling the basic requirements for photothermal agents used in biomedical applications [4].

In terms of power, to achieve the final product, through the Pechini method the reaction mixture has to pass through four different heating processes: dissolution of the lanthanide precursors at 353 K, evaporation of water at 373 K, precalcination process at 573 K, and final calcination at 1023 K. For the MW and CA solvothermal methodologies, only three steps are required: reaction at 453 K, solvent evaporation at 353 K and calcination at 1023 K. 

Lately, another microwave-assisted hydrothermal synthesis method for the preparation of monoclinic KY(WO_4_)_2_ nanocrystals without using organic surfactants, has been proposed [50]. This reaction involves preheating at 388 K for 0.5 h, a reaction at 523 K for 3 h, a purification process with deionized water, and a drying process at 353 K for 20 h, after calcination at 1023 K for 1 h. The final product of the reaction shows a tendency towards agglomeration and irregular morphological habits. Nevertheless, the ability of MW and CA solvothermal methods to generate discrete nanoparticles with no agglomerations, close-to-regular shapes and through a time and energy-effective method, offers significant advantages in the synthesis of these nanoparticles.

The two solvothermal based methodologies reported here, display differences among them, especially in the size and shape of the seeds and final products obtained. The seeds obtained by these two methods, exhibit a similar XRPD pattern (Appendix A), but a significant different morphology (Appendix A). The seeds obtained by the MW method are nanorods (Appendix A) with an average length of 322 nm, whereas the seeds obtained by the CA solvothermal method exhibit an urchin-like morphology (Appendix A) with an average length of 585 nm. The differences between these two approaches should be attributed to the different heating mechanisms to achieve the desired reaction temperature and the way in which the precursors behave exposed to these mechanisms, affecting hence, the kinetics of the reaction and as a consequence, the nucleation and growth mechanisms of the seeds. The MW solvothermal method drives chemical reactions by taking advantage of the ability of the used reagents and solvents to transform microwave irradiation into heat. Therefore, chemical reactions become more efficient and can be performed in a shorter time due to the selective absorption of microwave energy by polar molecules, in most of the cases, leading to an uniform heating in the reaction mixture [51]. The CA solvothermal reaction, from its side, involves the use of a furnace as the heating system, which heats the walls of the reactor by thermal convection or conduction, leading to longer times for the reaction mixture to reach a homogenous distribution of the desired temperature, and as a consequence the reaction mixture is not uniformly heated while part of the energy is wasted by dissipation [51].

### 3.2. Mechanism of the Formation of the KLuW Nanocrystals via the MW or CA Solvothermal Methodologies

Taking into account all the parameters that affect to the size, shape and formation of KLuW nanocrystals in the solvothermal synthesis, an insight to the possible formation mechanism is attempted. We believe that in the synthesis of the monoclinic KLuW nanocrystals takes place a so-called “cooperative-controlled crystallization mechanism”, which was introduced by Bu et al., for the synthesis of uniform bipyramidal tetragonal NaLa(MoO_4_)_2_ nanocrystals [36]. In this mechanism, the role of the organic surfactants was to control the nucleation and growth of NaLa(MoO_4_)_2_ nanocrystals by selectively binding to a specific crystal facet. The main difference between the synthesis of the two compounds is that while tetragonal NaLa(MoO_4_)_2_ can be obtained directly from the solvothermal reaction, monoclinic KLuW could not be obtained as a direct product of the MW or CA solvothermal method, but a calcination step was crucial to achieve the desired KLuW phase. Hence, here the cooperative-controlled crystallization mechanism would affect the formation of the seeds, having a critical role in defining the final size of the KLuW nanocrystals.

Scheme 1 summarizes the proposed possible formation mechanism of the monoclinic KLuW nanocrystals via the MW (or CA) solvothermal methods. When OLAC and OLAM are mixed in an equimolar ratio in ethanol, a mixture of deprotonated OLAC containing carboxylate anions (C_17_H_33_COO^−^, plotted in the scheme as OLAC^−^), protonated OLAM (C_18_H_35_NH_3_^+^, plotted in the scheme as OLAM^+^) and the corresponding acid-base complex (C_17_H_33_COO^−^:C_18_H_35_NH_3_^+^, plotted in the scheme as OLAM^+^ OLAC^−^) are present. In fact, as stated by Bu et al. OLAM helps enhancing the deprotonation of OLAC and forms an acid-base complex with OLAC molecules. OLAM also contributes to keep the pH values stable to 8.7 [36], which was confirmed experimentally in both MW and CA solvothermal methods.

By adding the Lu(NO_3_)_3_·H_2_O aqueous solution, a lutetium-OLAC complex, i.e., Lu(NO_3_)_3−x_(C_17_H_33_COO)_x_ would form via anion exchange. Proof of the complexation between the lanthanide cation and the OLAC anion is the change of the aspect of the solution from a homogeneous transparent solution to a milky one. After the addition of the aqueous solution of K_2_WO_4_, an ion exchange and substitution among NO_3_^−^, OLAC^−^ and WO_4_^2−^, as well as Lu^3+^ and K^+^ takes place, leading to the formation of the primitive nucleus precursors [36,52]. During the solvothermal reaction, the precursor’s nucleuses reach a critical size leading to the formation of stable nucleus of the precursor compound. Later on, solvothermal growth and ripening processes result in the production of the initial anisotropic carboxylate anions caped seeds. The enhanced deprotonation of OLAC, having a superior electron-donating ability than OLAM, facilitates the binding to metal ions on the facets of the seeds [53]. It is worth to mention that in the case of the MW reaction, this binding process seems to be more anisotropic, and the OLAC capping groups shows a preferential binding, observed by an enhanced XRPD intensity of the angle at around 10° (Appendix A) starting the formation of the nanorod morphology. Since the crystalline nature of the seeds could not be identified, it is not possible to identify to which crystal faces, these deprotonated OLAC molecules would bind preferentially. 

After the solvothermal treatment, the formed seeds coated with oleic acid, sediment at the bottom of the Teflon microwave or autoclave vial after centrifugation because of the effect of gravity and their hydrophobic surfaces. Free OLAC dimers, OLAM, protonated OLAM, acid-base complexes and the excess of organic surfactants would not bind to the seed surfaces and would be washed away during the purification steps with ethanol and hexane. Finally, to achieve the desired crystalline phase, the seeds were calcined at 1023 K for 2 h at ambient atmosphere. Apparently, the OLAC-OLAM system acts like a media-regulator, maintaining a stable pH value in the mixture, which is a key feature for the formation of seeds that will be later calcined to obtain the monoclinic KLuW phase. When excess of OLAM or no organic surfactants were used, the final product of the synthesis was a mixture of compounds (Appendix A) instead of pure KLuW. OLAM contributes to keep the pH values stable to 8.7 [36], which was confirmed experimentally in both MW and CA solvothermal methods. This value of the pH is highly important for the formation of the desired seed precursors and pure monoclinic phase. Thus, tuning the value of the pH would greatly influence the degree of the hydrolysis and the state of the cations and tungsten ions present in this liquid phase chemical reaction. Since the organic surfactants maintain these stable pH values, upon removing them (reaction MW-10), the pH value dropped to 7.55, as a consequence the XRPD patterns of the seeds and the corresponding final product is highly different that the reactions with pH 8.7 (Appendix A). At pH 7.55, the final product obtained was a mixture between K_2_WO_4_ and Lu_2_O_3_ (Appendix A). On the other hand, the increase of the value of the pH to 10.2, when excess of OLAM (4 times) in comparison with OLAC was added (reaction MW-9), additional low intensity peaks are observed in the XRPD pattern of the final product (Appendix A).

Further surface FT-IR characterization of the dried seeds (Appendix A) confirms that they are coated with oleic acid. These results confirm the adsorption of the oleyl group on their surfaces, by the presence of the peaks related to the -CH_2_ asymmetric and symmetric stretching vibrations of this group, observed at 2930 cm^−1^ and 2850 cm^−1^, respectively [41,54]. The characteristic peaks of OLAC (1709 cm^−1^) and OLAM (1593 and 3300 cm^−1^) could not be identified in the spectrum, indicating that no free OLAC or OLAM are present on the samples [55]. The broad band observed at 3440 cm^−1^, can be attributed to the stretching vibration modes of the O-H bonds, generated by the adsorption of oleic acid on the surface of the seeds. Finally, the two strong peaks observed at 1625 and 1460 cm^−1^ indicate the bidentate coordination of OLAC to the metal atoms [56]. OLAM is not observed on the FT-IR spectrum of the seeds, confirming that OLAM helps to the formation of an acid-base complex with OLAC, and consequently, more OLAC molecules present dissociated hydroxyl groups that facilitate the adsorption of OLAC on the surface of the seeds. This observation is in agreement with other papers published previously [41,57], and also with the case when only OLAM was used as organic surfactant [58]. After calcination at 1023 K, there is no presence of the capping surfactants, as confirmed by the FT-IR spectrum corresponding to the KLuW nanocrystals (Appendix A), and also by the DTA results (Figure 1d). Another factor that confirms the successful elimination of the organic coating is the dispersibility of the final nanocrystals in water (Appendix A). If the hydrophobic organic surfactants would be present on the surface of the KLuW nanocrystals, they would not be dispersible in water due to their hydrophobic nature.

### 3.3. Photoluminescence Characterizations

The emissions of the Ho, Tm:KLuW nanoparticles synthesized via MW and CA solvothermal methodologies, lying in the third biological window (III-BW), were recorded at room temperature after excitation at 808 nm with a power of 200 mW. The emission spectra, regardless of the doping ratio applied, consist of three emission bands: 1.45 µm and 1.8 µm, corresponding to the ^3^H_4_ → ^3^F_4_ and ^3^F_4_ → ^3^H_6_ electronic transitions of Tm^3+^, and 1.96 µm, corresponding to the ^5^I_7_ → ^5^I_8_ electronic transition of Ho^3+^ (Figure 4a for 3 mol% Ho, 5 mol% Tm:KLuW and Appendix A for 1 mol% Ho, 10 mol% Tm:KLuW), respectively [36]. These doping ratios were selected according to a previous work achieved from our group in which these doping ratios, among others, exhibited the highest photoluminescence intensity in the III-BW (3 mol% Ho^3+^, 5 mol% Tm^3+^) and the highest photothermal conversion efficiency (1 mol% Ho^3+^, 10 mol% Tm^3+^) [29]. Comparing the intensity of the emission generated, the nanocrystals synthesized by the MW method, produce a brighter emission.

The mechanisms of generation of these emission bands is depicted in Figure 4b. Tm^3+^ absorbs a photon at 808 nm and promotes its electrons from the ^3^H_6_ ground state to the ^3^H_4_ excited state. From there, the electrons decay radiatively to the ^3^F_4_ manifold, generating the emission line at 1.45 μm. With another radiative decay from the ^3^F_4_ level to the ^3^H_6_ ground state, the emission line at 1.8 μm is generated. Tm^3+^ ions also undergoes cross-relaxation (CR) process into the ^3^F_4_ excited state, when one of the ions is initially excited into the upper ^3^H_4_ excited state and then it relaxes non-radiatively to the ^3^F_4_ level while this energy is used to promote an electron in the ^3^H_6_ ground state to the ^3^F_4_ level, due to the energy resonance between these two processes. Due that the ^3^F_4_ level of Tm^3+^ and the ^5^I_7_ level of Ho^3+^ are resonant in energy, energy transfer (ET) and back energy transfer (BET) processes might take place, promoting the electrons of Ho^3+^ from the ground state to this excited state. Then, the electrons of Ho^3+^ relax back radiatively to the ^5^I_8_ ground state, giving rise to the emission band at 1.96 μm [29]. 

### 3.4. Ho, Tm:KLuW Nanocrystals as Luminescent Thermal Sensors in the Third Biological Window

The thermal sensing capacity in the III-BW spectral region of Ho, Tm doped KLuW nanoparticles synthesized via MW and CA solvothermal methodologies for the doping concentration of 1 mol% Ho, 10 mol% Tm and 3 mol% Ho, 5 mol% Tm, has been evaluated and compared to the performance of other thermal sensors, by studying the dependence of the intensity ratio (Δ), the absolute (S_abs_) and relative (S_rel_) thermal sensitivities, and the temperature resolution (δT), in the physiological range of temperatures between 293 K and 333 K after 808 nm excitation using 200 mW laser power.

With the increase of the temperature, the intensity of the Ho^3+^ emission band decreases (Figure 5a) due to the thermal activation of luminescence quenching mechanisms [27], while the intensity of the Tm^3+^ emission bands remains almost unchanged, hence they can be used as a reference for thermal sensing.

The temperature dependence of the intensity ratio, can be modelled through the approach for dual center emission lanthanide doped thermometers [59]. This model (a full derivation of this model is explained in Appendix A) is based on the fact that the total transition probability of a particular emitting level is the sum of the radiative and non-radiative transition probabilities [60], and the integrated luminescence intensity can be correlated to the inverse of the total transition probability [61]. The dependence of the intensity ratio vs. temperature can be expressed by:(1)$$\Delta =\frac{I_{1}}{I_{2}}=\Delta_{0}\frac{1+\sum_{i}\alpha_{2i}\exp\left(-\frac{\Delta E_{2i}}{k_{B\;}T}\right)}{1+\sum_{i}\alpha_{1i}\exp\left(-\frac{\Delta E_{1i}}{k_{B\;}T}\right)}$$
where 1 and 2 are the two emissions whose intensities are used to estimate the thermometric performance; Δ_0_ stands for the ratio between the I_01_/I_02_ at 0 K for the 1 and 2 emissions; α_2i_ and α_1_ stands for the ratio between the non-radiative and radiative probabilities for the emitting level of the electronic transitions 1 and 2, respectively; and the sum sign extends from I = 1 to n, being n all the possible non-radiative deactivation channels of the electronic starting levels for the transitions with intensities I_1_ and I_2_. Finally, ΔE_2i_ and ΔE_1i_ are the activation energies for the thermally quenched processes of the transitions 1 and 2, and k_B_ is the Boltzmann constant expressed in cm^−1^ (k_B_ = 0.695 cm^−1^). 

If the exponential term dominates in the intensities of the transitions involved and assuming a single deactivation channel (1<<α_j_ exp (−ΔE_j_/k_B_T), Equation (1) could be converted to:(2)$$\Delta =\Delta_{0}\frac{\alpha_{2i}}{\alpha_{1i}}\;\frac{\exp\left(-\frac{\Delta E_{2}}{k_{B\;}T}\right)}{\exp\left(-\frac{\Delta E_{1}}{k_{B\;}T}\right)}=B\exp(\frac{\Delta E_{1}-\Delta E_{2}}{k_{B}T})=B\exp(\frac{-C}{T})$$
where B =$\;\Delta_{0}\frac{\alpha_{2i}}{\alpha_{1i}}$ is an empirical constant to be determined by experimental fitting, and $C=\frac{\Delta E_{1}-\Delta E_{2}}{k_{B}}$ is the energy difference between the two activation energies for the thermally quenched processes divided by the Boltzmann’s constant.

Compared to the other two intensity ratios that can be calculated (the 1.45 μm/1.8 μm and 1.45 μm/1.96 μm), the intensity ratio between the bands at 1.8 μm and 1.96 μm, is the one that results affected the most by the increase of temperature (Figure 5b and Appendix A). This is in agreement with the results obtained previously for Ho, Tm:KLuW nanoparticles synthesized by the modified sol-gel Pechini method [29]. For this intensity ratio, by fitting Equation (2) to the experimental emission-temperature dependence for each Ho, Tm:KLuW sample (Figure 5c), we determined the values of B and C. The results are summarized in Appendix A. In addition, we explored the power dependence of the intensity ratio for the three emission bands. This relation is of paramount interest when investigating the application of these lanthanide doped luminescent thermometers in biomedical fields [62]. This relation is linear: with the increase of the excitation power, the intensity ratio increases (Figure 5d). The intensity ratio that exhibits the highest change with the increase of the excitation power is the one between the emissions located at 1.8 μm and 1.96 μm [29]. This linear relation is important during biological application of these lanthanide doped luminescent thermometers where is no real control on the power reaching the particles and this might rise imprecision during measurements. Thus, having a linear relation, this imprecision source is reduced.

Knowing the value of B and C for each Ho, Tm doped KLuW materials, we can estimate their thermometric performance by calculating S_abs_, S_rel_ and δT. The absolute thermal sensitivity depends on the experimental setup used to record the spectra and the characteristics of the sample (absorption and lifetimes) [59]. Hence, S_abs_ will allow the comparison of only Ho, Tm: KLuW analyzed using the experimental setup applied here, but not allow the comparison with different classes of thermometers. The absolute thermal sensitivity is expressed by [59]:(3)$${\;S}_{\mathrm{abs}}=\frac{\partial \;\Delta }{\partial \;T}\;$$

The samples with the highest absolute thermal sensitivity are the ones synthesized by the MW solvothermal methodology, with 1 mol% Ho and 10 mol% Tm dopant concentrations (Figure 6a and Appendix A). 

To compare different classes of thermometers, the relative thermal sensitivity is used as a Figure of merit and expresses the maximum change in the intensity ratio for each temperature degree [59]. The relative thermal sensitivity is defined as [59]:(4)$${\;S}_{\mathrm{rel}}=\;\frac{1}{\Delta }\;\left|\frac{\partial \Delta }{\partial T}\right|\;\times \;100\%=\left|\frac{\Delta E}{k_{B}T^{2}}\right|\;\times \;100{\%}^{\;}$$
where $\Delta E=\Delta E_{2}-\Delta E_{1}$ is determined by experimental fitting and T is the absolute temperature. 

In general, the intensity of the emissions generated by the Ho, Tm:KLuW nanocrystals synthesized by the CA solvothermal method is more sensitive to the changes of temperature when compared to the nanocrystals synthesized by the MW method. The 3 mol% Ho, 5 mol% Tm doping concentrations showed the highest relative thermal sensitivity with a value of 0.33% K^−1^ at room temperature (Figure 6b and Appendix A). The lowest S_rel_ is obtained for the 1 mol% Ho, 10 mol% Tm: KLuW nanocrystals synthesized by the MW-assisted methodology. A general trend in the temperature dependence of the relative thermal sensitivity with the temperature is its decrease as the temperature increased (Figure 6b).

The temperature resolution is defined as the smallest temperature change that can be detected in a given measurement and it is estimated by [59]:(5)$$\delta T=\;\frac{1}{S_{\mathrm{rel}}}\;\frac{\delta \Delta }{\Delta }=\left|\frac{k_{B}T^{2}}{\Delta E}\right|\;\frac{\delta \Delta }{\Delta }\;$$
where $\frac{\delta \Delta }{\Delta }\;$= 0.5% is the relative error in the determination of the thermometric parameter, determined by the acquisition setup used [28]. The variation of the temperature resolution with the increase of the temperature follows the inverse trend of the relative thermal sensitivity. Hence, the better is the relative thermal sensitivity, the lowest is the temperature resolution (Figure 6c and Appendix A). The lowest δT corresponds to 3 mol% Ho, 5 mol% Tm: KLuW synthesized by the CA methodology with a value of 1.5 K at 293 K. As the temperature increases, this value increases up to a maximum of 1.94 K at 333 K.

Furthermore, we compared the thermometric performance of these nanocrystals with the corresponding Ho, Tm:KLuW synthesized from the modified sol-gel Pechini methodology. Hence, Ho, Tm:KLuW from this methodology exhibit higher thermometric performance compared to the solvothermal methodologies. Being recorded with the exact same experimental setup, the S_abs_ in both doping ratios from the modified sol-gel Pechini methodology are relative high compared to the corresponding ratios from the MW and CA solvothermal methodology (Figure 6a). The trend is furthermore extended also for the case of the relative thermal sensitivity (Figure 6b). It should be noted here that the doping ratio 1 mol% Ho and 10 mol% Tm:KLuW synthesized by the modified sol-gel Pechini methodology, displays one of the most sensitive thermometers operating in the III-BW [29]. The temperature resolution, being reverse proportional to the relative thermal sensitivity, exhibited the lowest value for the product of the sol-gel Pechini methodology (Figure 6c). Apparently, the thermometric performance of Ho, Tm:KLuW nanocrystals decreases with the decrease of their sizes. So, Ho, Tm:KLuW sensors synthesized by the modified sol-gel Pechini methodology, CA and MW, have sizes ranging from 150 nm [29], 16 nm and 12 nm, respectively, and their corresponding maximum S_rel_ are in the range of 0.90% K^−1^, 0.33% K^−1^ and 0.23% K^−1^, respectively. Similar results are obtained in the case of Er^3+^, Yb^3+^:NaYF_4_ [63], Er^3+^:Y_2_O_3_ [64], Pr^3+^:LaF_3_ and Pr^3+^:LiYF_4_ sensors [65]. It was argued that as the size of the particles decreases, the amount of the luminescent active ions close to their surfaces is increased due to higher surface to volume ratio. These ions can interact with the ligands attached to the surface of the nanoparticles, leading to quenching processes that would affect negatively their thermal sensitivity [63,64].

Having determined S_rel_ and δT, now we can compare the performance of these nanocrystals with other sensors operating in the III-BW. The performance of 1 mol% Ho and 10 mol% Tm:KLuW synthesized by MW and CA is comparable to that of Tm, Yb:KLuW [66] and Er, Yb:LuVO_4_@SiO_2_ [67] (Figure 6d). Nevertheless, the performance of MW and CA nanocrystals is lower compared to the Tm, Yb, Ho:KLuW [66] and Tm, Yb:NaYF_4_ [66]. 

### 3.5. Ho, Tm:KLuW Nanocrystals as Photothermal Conversion Agents

When a material is illuminated with light, it can be absorbed, scattered, reflected and/or transmitted. The part of the light which is absorbed can be converted in photoluminescence and heat. In lanthanide doped materials, this transformation into heat is due to the non-radiative processes that can occur in the relaxation of the electrons of the ion. The photothermal conversion efficiency, i.e., the ability of the nanocrystals to convert the absorbed light into heat, was determined using the integrating sphere methodology [37].

In this method, the photothermal conversion efficiency (η) is calculated from the expression:(6)$$\eta =\left|\frac{P_{\mathrm{blank}}-{\;P}_{\mathrm{sample}}}{P_{\mathrm{empty}}-P_{\mathrm{sample}}}\right|\times 100\%$$where P_blank_, P_empty_ and P_sample_ are the power values measured for the solvent (distilled water in this case), the empty sample holder and the doped Ho, Tm:KLuW nanoparticles synthesized via the solvothermal methodologies, respectively. 

The photothermal conversion efficiency of the Ho, Tm:KLu(WO_4_)_2_ nanoparticles was measured by illuminating a sample of the Ho, Tm:KLu(WO_4_)_2_ nanoparticles dispersed in distilled water with a concentration of 1 g L^−1^ with a laser emitting at 808 nm with a power of 200 mW. Figure 7 summarizes the photothermal conversion efficiencies obtained. The highest light-to-heat conversion efficiency was obtained by the nanoparticles with the smallest sizes, synthesized by the MW solvothermal method with a value of η = 45 ± 2% for the 1 mol% Ho, 10 mol% Tm doped KLuW nanoparticles. The same material, synthesized by the CA approach and with slightly larger sizes, displays a light-to-heat conversion efficiency slightly lower, of 43 ± 3%. Instead, when the doping levels were 3 mol% Ho, 5 mol% Tm, the highest photothermal conversion efficiency was obtained for the nanocrystals synthesized by the CA methodology with a value of 36 ± 3%. This is not surprising since the corresponding 3 mol% Ho, 5 mol% Tm KLuW nanocrystals synthesized by the MW-assisted method present the highest emission intensity among all the nanocrystals analyzed (Figure 4a). 

In addition, we explored the excitation power dependence of the photothermal conversion efficiency. The results obtained reveal that the change of the pumping power has no influence on the photothermal conversion efficiency (Figure 7). The increase of the concentration of Tm^3+^ favors the increase of the probability of multiphonon decay transitions, which in turn favors the generation of heat [29]. Also, it seems that the decrease of size of the material enhances the non-radiative processes to happen, which are responsible for the generation of heat. If we compare the morphology of these nanodimensional Ho, Tm:KLuW particles (see Figure 3), it can be noted that they tend to form agglomerates. However, small nanoparticles in the volume are located in the form of one-dimensional agglomerates, while larger ones are located in the form of three-dimensional agglomerates. Therefore, although the CA methodology produces larger particles, which should generate more heat under equal pumping conditions, as the magnitude of heat flux into the environment is proportional to the volume of the nanoparticles [68,69], the strong agglomeration observed in this system does not allow an effective use of their more advanced thermo-physical properties (large surface and volume) for heating the environment. On the contrary, small particles in solution in the form of branched filaments are much less agglomerated, therefore have better contact with the environment, which leads to higher heat transfer efficiency. This conclusion can be confirmed furthermore if we compare also the results obtained from the particles synthesized by the modified sol-gel Pechini method (see Figure 7). These particles tend to form big agglomerates, bigger than those of the nanoparticles synthesized via the solvothermal methodologies, leading to smaller heat efficiencies. 

Compared to other classes of photothermal conversion agents, the Ho, Tm:KLuW nanocrystals synthesized by solvothermal methods exhibit a higher photothermal conversion efficiency than FePt nanoparticles [70], Cu_9_S_5_ [22], graphene oxide [37], Au/AuS or Au/SiO_2_ nanoshells (Table 1) [71]. The photothermal conversion efficiency reported here is of the same order of magnitude than that reported for graphene in DMF [37] and Au nanoshells or Au nanorods [10]. However, the ability of Au nanostars [10], NaNdF_4_ [72], NdVO_4_ in water [73], or core@shell@shell NaNdF_4_@NaYF_4_@Nd:NaYF_4_ [74] to generate heat is still higher compared to the Ho, Tm doped KLuW nanocrystals reported here (Table 1).

### 3.6. Ho, Tm:KLuW Nanocrystals as Self-Assessed Photothermal Agents

Ho, Tm:KLuW nanocrystals exhibit the ability to self-determine the temperature achieved by the system when releasing heat by using luminescence thermometry, generating self-assessed photothermal agents. This ability was proved by monitoring the temperature generated by a dispersion of these nanoparticles with a concentration of 1 g L^−1^, after excitation at 808 nm with a power of 200 mW and a beam spot diameter of approximately 10 μm on the external surface of the vial, while recording the photoluminescence spectrum. 

Figure 8a illustrates the temperature evolution in the water dispersion of the Ho, Tm doped KLuW nanocrystals synthesized by the solvothermal methodologies. The nanocrystals with a doping concentration of 1 mol% Ho and 10 mol% Tm allows achieving the maximum temperature of 316.8 K in 100 s. In addition, the temperature evolution of pure distilled water is included for comparison. From the data it can be concluded that the temperature increase of more than 20 K achieved has to be attributed to the light-to-heat conversion ability of these nanocrystals. The general tendency observed, independently of the doping concentration in the nanocrystals, is a fast increase of the temperature in the first 25–30 s and then a slower increase of temperature until reaching the thermal equilibrium after approximately 100 s, in which, despite the water dispersion of nanocrystals is still illuminated with the 800 nm laser, the temperature does not increase further. Further, we compared also the temperature profile of modified sol-gel Pechini methodology, and the increase of the temperature are in accordance with the data obtained from the photothermal conversion efficiency (Figure 8a). Results reconfirm that the smaller and non-aggregated nanoparticles synthesized from the solvothermal methodologies, have higher temperature increase compared to the Pechini ones. This trend is observed for both doping ratios studied (1 mol% Ho, 10 mol% Tm and 3 mol% Ho, 5 mol% Tm).

Further, we tested the sedimentation time of the nanocrystals synthesized from these different morphologies. It is generally accepted that large and heavy nanoparticles can sediment quickly, causing the dose of nanoparticles on biological applications to vary [76,77]. In such cases, the actual concentration of nanoparticles in biomedical systems, could be significantly different from the initial provided value. And since, the reliability of these biological systems is strictly related to the concentration of the nanoparticles, these variations in the concentration of nanoparticles, may be a source of inaccuracies and misleading [76,77]. To test sedimentation properties, the nanocrystals with a concentration of 1 g/L were dispersed in distilled water, and different photographs of the vials were acquired as function of the time (Appendix A). It can be observed that the nanocrystals synthesized from the solvothermal methodology (MW approach as an illustrative example) are stable for longer times compared to the aggregated nanoparticles achieved from the modified sol-gel Pechini methodology. Precisely speaking, the modified sol-gel Pechini methodology start to sediment at the bottom of the vial after 2 h, whereas the MW solvothermal methodology are stable up to 4.5 h. These results indicate that the nanocrystals produced from the solvothermal methodologies are more stable in biological compatible fluids, such as distilled water, which increases their potential to be applied for temperature determination in biomedical applications.

To prove the concept of the self-assessed photothermal agents in real biological samples in an ex-vivo experiment, 1 mol% Ho and 10 mol% Tm:KLuW materials synthesized via MW assisted solvothermal method (the particles with the highest photothermal conversion efficiency) were injected into a chicken breast piece of meat and the temperature reached was measured. When illuminated with the 808 nm light, the nanoparticles will generate photoluminescence in the III-BW, simultaneously, heat will be also released [29]. Hence, the goal of the self-assessing properties of these nanocrystals is to use the generated photoluminescence to determine how much the temperature increases inside the chicken breast piece of meat. This temperature is also measured by a thermocouple located near the injected nanoparticles to verify that the temperature determined by luminescence means is correct. In addition, we implemented to the optical setup an infrared camera with the goal to record the temperature of tissue’s surface. In this way, we monitor the temperature inside the tissue and also at the surface of the chicken breast. As can be noted from Figure 8b, the temperature recorded at these two positions is remarkable different. The temperature measured inside the chicken breast from the thermocouple is 313.4 K, whereas the one at the surface of the tissue is around 305.2 K, leading to a difference of 9.2 K (22%) among them. These large differences are in accordance with other reports dealing with ex-vivo and in-vivo photothermal therapies based on heating nanoparticles [78,79,80] and are assigned to the heat diffusion processes taking place at the surface of the tissue [78,79,80].

Hence, having established that the proper temperature read-out is achieved inside the chicken breast, we placed around 25 mg of the nanoparticles and a thermocouple, in between two pieces of chicken breast, 2 mm thick, ensuring the same medium in all directions. The nanoparticles were then irradiated with the 808 nm laser, using a power of 200 mW and a spot size of ~10 μm on the surface of the chicken breast. This value of the power of the laser was the maximum power showing no degradation or burning of the surface of the chicken breast. It has to be taken into account that the power of the beam reaching the Ho, Tm:KLuW particles inside the chicken breast is reduced by the scattering and the absorption of the biological tissue. The luminescence generated by the nanoparticles was recorded when the temperature measured by the thermocouple was stabilized (after around 100–120 s from the beginning of the experiment). From this spectrum, the 1.8 μm/1.96 μm intensity ratio could be extracted and by using the calibration curve shown in Figure 5c the temperature inside the chicken breast was determinate. This temperature was then compared with the one measured by the thermocouple located close to the nanoparticles. Figure 8c presents the III-BW emissions generated by the nanocrystals, transmitted through the 2 mm thick chicken breast, together with the spectrum of the same nanoparticles not covered by the chicken breast, recorded in open air, showing that the biological tissue does not affect the measurements of the temperature by attenuating one of these emission bands more than the others.

The temperature determined by the luminescent thermometer was 313.1 K, whereas the thermocouple indicated a temperature of 313.4 K (Appendix A). Hence, the difference between the temperature determined by the luminescent thermometer and the thermocouple is only 0.3 K. Typical temperature difference was reported also in the case of the implementation of Ho, Tm doped KLuW nanocrystals synthesized via the modified sol-gel Pechini methodology, [29] however in this case, the difference is slightly lower. In addition, for the Pechini method, the maximum temperature. For the modified sol-gel Pechini methodology, the difference in temperature is in the range of 0.7 K [29]. The smaller difference exhibited by the products of the solvothermal methodology might rely to their smaller size and shape which ensures better distribution of the nanoparticles in the medium. Possible explanations for the small difference between the temperature determined by the luminescence thermometer and the thermocouple might rely on: (i) the different thermal conductivity of the dielectric material and the metallic thermocouple, and (ii) the fact that the calibration of the luminescent thermometer was performed in air and not directly inside the biological tissue. In fact, it has been reported that the medium in which the nanoparticles are embedded highly affects the determination of the temperature by luminescent thermometry [81,82,83]. However, the small difference observed shows that in preliminary experiments it is not necessary to perform this calibration if care is taken to verify that the thermal equilibrium has been reached.

The technique presented here demonstrates the potential applicability of Ho, Tm doped KLuW nanocrystals as multifunctional materials which can act simultaneously as luminescent nanothermeters operating in the III-BW and as photothermal agents, regardless of their size and shape. Furthermore, compared to the traditional modified sol-gel Pechini methodology, the products of the solvothermal methodology offer smaller sizes and no agglomeration, which in turn results in no sedimentation for longer times compared to agglomerated Ho, Tm:KLuW synthesized from Pechini methodology. The sedimentation data reveal that Ho, Tm doped KLuW nanocrystals achieved from solvothermal methodologies can be dispersed for longer times in solvents for temperature determination in biomedical applications. In addition, as a benefit of their smaller sizes, a lower temperature difference recorded from the nanocrystals and an external thermocouple in an ex-vivo experiment, is determined.

## 4. Conclusions

Monoclinic KLu(WO_4_)_2_ nanocrystals with sizes below 20 nm were synthesized via microwave-assisted and conventional autoclave solvothermal methods, in the presence of organic surfactants (oleic acid and oleylamine). The formation of the monoclinic crystalline phase was confirmed from X-ray powder diffraction pattern, Raman vibrational modes and differential thermal characterizations. These solvothermal methodologies introduced here, offer final product with defined shape and no agglomeration, in addition they are time and energy-effective when compared to other methods of synthesizing monoclinic KLu(WO_4_)_2_ particles, such as the modified sol-gel Pechini method. These two methodologies may provide some guidance in the preparation of other type of nanomaterials including vanadates, carbonates, molybdates and phosphates. 

These nanocrystals, act as self-assessed photothermal agents in the third biological window, upon doping with Ho^3+^ and Tm^3+^ ions and excitation with near infrared light, such as the 808 nm, lying in the first biological window. These doped nanocrystals can simultaneously generate luminescence and heat, which are the sources of their applications as nanothermometers and photothermal agents, respectively. The III-BW emissions located at 1.45 μm, 1.8 μm and 1.96 μm, are attributed to the ^3^H_4_ → ^3^F_4_ and ^3^F_4_ → ^3^H_6_ electronic transition of Tm^3+^ and ^5^I_7_ → ^5^I_8_ electronic transition of Ho^3+^. Upon doping with 1 mol% Ho, 10 mol% Tm and 3 mol% Ho, 5 mol% Tm for both microwave-assisted and conventional autoclave synthesis, the highest thermometric performance among these nanothermometers, studied at the physiological range of temperature (293–333 K), was assigned to the 3 mol% Ho, 5 mol% Tm nanoparticles synthesized from the conventional autoclave method, for a value of the relative thermal sensitivity of 0.33% K^−1^ and temperature resolution of 1.4 K at room temperature. Related to their photothermal conversion efficiency, the 1 mol% Ho, 10 mol% Tm nanoparticles synthesized from the microwave-assisted method, exhibit the highest efficiency for a value of 45 ± 2%. These results underline that the smaller size particles obtained generated heat more efficiently, but their capacity to sense temperature was not as good as that of the agglomerated nanocrystals achieved by the modified sol-gel Pechini methodology, revealing that the synthesis method influences the performance of these self-assessed photothermal agents. In addition, these nanocrystals were probed as self-assessed photothermal agents: the same nanocrystals, upon excitation, released heat that will increase the temperature of the medium in which they are embedded, and luminescence, which allows the determination of the temperature in situ without the addition of an external thermal probe. The Ho, Tm:KLuW nanocrystals achieved temperature reading at a penetration depth of 2 mm inside a chicken breast.

## Data Availability

The data that support the findings of this study are available from the corresponding author, upon reasonable request.

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
