# Peer review of "Effect of the Size and Shape of Ho, Tm:KLu(WO_4_)_2_ Nanoparticles on Their Self-Assessed Photothermal Properties"

_nanomaterials, 2021, doi:10.3390/nano11020485_

Round 1
Reviewer 1 Report
The authors investigated the effect of the size and shape of Ho, Tm:KLu(WO4)2 nanoparticles in their self-assessed photothermal properties. Although it is interesting, many parts (even sentences) are the exactly same as the published paper below:
"Short-wavelength infrared self-assessed photothermal agents based on Ho,Tm:KLu(WO4)2 nanocrystals operating in the third biological window (1.45–1.96 μm wavelength range)" published in J. Mater. Chem. C, 2020, 8, 180-191.
Author Response
The main purpose of the present manuscript was to present new synthesis methods for KLu(WO4)2 nanoparticles, based on solvothermal approaches. This, as highlighted in the Academic Editor Notes facilitated together with the comments of the reviewers, this by itself is quite important. In fact, it is the first time that real discrete nanoparticles of KLu(WO4)2 nanoparticles have been produced, and given the technological importance of this material, especially for the development of lasers, open the door to the investigation of new applications, such might be the preparation of nanoceramic samples of this material.
To enrich the manuscript we decided to include an application for the nanoparticles newly prepared, and we decided to include the application in which our laboratory is working at present, that is self-assessed photothermal agents. Thus, in this way, at the same time we have been able to compare the performance of the nanoparticles prepared by the solvothermal methods with those of the same material prepared by the Pechini method, investigating if the preparation method has any influence on the later properties of the material. We introduced this idea in the abstract, as well as in the conclusions, to make it more clear the purpose of the manuscript:
Abstract: “The self-assessing properties of these nanocrystals are studied as a function of their size and shape and compared to the ones prepared by the modified sol-gel Pechini methodology, revealing that the small nanocrystals obtained by the hydrothermal methods have the ability to generate heat more efficiently, but their capacity to sense temperature is not as good as that of the nanoparticles prepared by the modified sol-gel Pechnini method, revealing that the synthesis method influences the performance of these self-assessed photothermal agents.”
Conclusions: “These results underline that the smaller size particles obtained generated heat more efficiently, but their capacity to sense temperature was not as good as that of the agglomerated nanocrystals achieved by the modified sol-gel Pechini methodology, revealing that the synthesis method influences the performance of these self-assessed photothermal agents.”
Thus, the parts that are the same as the paper published in J. Mater. Chem. C, 2020, 8, 180-191 correspond to those of the description of XRD, TEM, photoluminescence and photothermal conversion efficiency, that obviously have been performed in the same way in the two papers, and that allowed as to make a full comprehensive understanding of the effect of these nanodimensional particles compared to the aggregated particles that appear in our previous article. To do that, it is crucial to have the same experimental setup to evaluate, for example, their photoluminescence in the third biological window, their nanothermometric and photothermal conversion applications. Furthermore, as considered in the Academic Editor Notes, the investigations in the present manuscript are much more complex than in the previous paper, which give the value and purpose of the present manuscript, and at the same time they are nanodimensional and biocompatible, thus fulfilling the conditions of being appropriately applied in biomedical applications, if considered.
Reviewer 2 Report
The manuscript presents the potential applicability of Ho, Tm doped KLuW nanocrystals as multifunctional materials which can act simultaneously as luminescent nanothermeters operating in the III-BW and as photothermal agents, regardless of their size and 808 shape. The topic is worth exploring. However, before publication the authors must analyze their experimental results more carefully and address several remarks.
1) Abstract “revealing that small nanocrystals generate more heat and sense the temperature less”
This is contrary to previous experiment and theory when larger nanocrystals produce more heat. [S. G. Fedorenko, et. al, Heating and cooling transients in the DyPO4 nanocrystals under femtosecond laser irradiation in the NIR spectral range, Physics of Wave Phenomena, 26, No. 3 (2018) 198–206, ISSN 1541-308X DOI: 10.3103/S1541308X18030044]
2) Please, give the derivation of Eq. (1) and (2) in Supplement, because the references are unavailable. Now, it is impossible to check the correctness of the idea for temperature measurement and the equations.
3) It is written in p. 18 lines 674 – 675 “Up to date, there are no reports concerning a size and shape dependence study on the photothermal conversion efficiency of lanthanide doped materials.” This is not entirely true. There is a work on the dependence of the efficiency of photothermal conversion of materials doped with lanthanides on the NP size. See the link above. I suggest looking more for journals with less impact factor, but with more physics.
4) Figure: 9. Temperature of 315 ° C is insufficient for medical treatment.
5) The presence of organic surfactants (oleic acid and oleylamine) does not allow the use of NPs for real medical purposes due to their toxicity. Please, comment.
6) The article turns out to be too long, but at the same time there is no necessary in-depth analysis of the experimental results.
Author Response
1) Abstract “revealing that small nanocrystals generate more heat and sense the temperature less” This is contrary to previous experiment and theory when larger nanocrystals produce more heat. [S. G. Fedorenko, et. al, Heating and cooling transients in the DyPO4 nanocrystals under femtosecond laser irradiation in the NIR spectral range, Physics of Wave Phenomena, 26, No. 3 (2018) 198–206, ISSN 1541-308X DOI:10.3103/S1541308X18030044]
Reply:
The referee is right. Strictly speaking and as indicated by the mentioned reference, the heat flux radiated is larger as larger is the particle. However, our intention was to describe the experimental observation that in the present work, the small nanocrystals obtained by the hydrothermal methods generate heat more efficiently when compared to the previous ones, prepared by the Pechini method, which were bigger. The reason for that is complex and should be related to many factors, but mainly related to the fact that the probability of the non-radiative processes to happen is favoured. Factors that might influence these processes are the doping levels used (see for example our previous paper in which different doping levels lead to different thermal sensing or heat generation properties: J. Mater. Chem. C, 2020, 8, 180-191), the amount of impurities present in the structures, or defects that might be generated during the synthetic methodologies. About the last two factors, we do not have control over them, which makes the prediction of the amount of heat generated quite complex. In addition, a significant effect is the flocculation time of the particles within the aqueous dispersion. Clearly, the smaller particles (the ones synthesized from the solvothermal methodologies) are more stable in an aqueous dispersion than the bigger ones, as it is confirmed by Figure S21 of the Supporting Information of our manuscript, which might imply that they are exposed to a longer irradiation time when compared to the bigger ones (the ones prepared by the sol-gel Pechini method).
So, taking into account this discussion, we changed the sentence in the abstract as:
“The self-assessing properties of these nanocrystals are studied as a function of their size and shape and compared to the ones prepared by the modified sol-gel Pechini methodology, revealing that the small nanocrystals obtained by the hydrothermal methods have the ability to generate heat more efficiently, but their capacity to sense temperature is not as good as that of the nanoparticles prepared by the modified sol-gel Pechnini method, revealing that the synthesis method influences the performance of these self-assessed photothermal agents.”
The same idea was expressed in the conclusions of the manuscript:
“These results underline that the smaller size particles obtained generated heat more efficiently, but their capacity to sense temperature was not as good as that of the agglomerated nanocrystals achieved by the modified sol-gel Pechini methodology, revealing that the synthesis method influences the performance of these self-assessed photothermal agents.”
2) Please, give the derivation of Eq. (1) and (2) in Supplement, because the references are unavailable. Now, it is impossible to check the correctness of the idea for temperature measurement and the equations.
Reply: We added in the Supporting Information, the derivation of Eq. (1) and Eq. (2) and a full list of the references supporting the theory behind the model estimating the temperature dependence of the thermometric parameter ∆. We kindly ask the reviewer to check Section 4. Modelling the temperature dependence of the thermometric parameter in the Supporting Information since equations could not be pasted properly in this reply.
3) It is written in p. 18 lines 674 – 675 “Up to date, there are no reports concerning a size and shape dependence study on the photothermal conversion efficiency of lanthanide doped materials.” This is not entirely true. There is a work on the dependence of the efficiency of photothermal conversion of materials doped with lanthanides on the NP size. See the link above. I suggest looking more for journals with less impact factor, but with more physics.
Reply: Taking into account the reference provided and the suggestion from the referee, we could find in literature some other articles that investigate the size-dependence of the photothermal conversion efficiency in lanthanide doped particles. As stated also from the reference [S. G. Fedorenko, et. al, Heating and cooling transients in the DyPO4 nanocrystals under femtosecond laser irradiation in the NIR spectral range, Physics of Wave Phenomena, 26, No. 3 (2018) 198–206, ISSN 1541-308X DOI: 10.3103/S1541308X18030044], they emphasize that the heat generated by the particles is proportional to their volume, larger particles heat the environment much more efficiently than smaller ones. However, small particles have a shorter relaxation time, and they heat up faster (albeit to a lower temperature) and cool faster, allowing the inertia of the hyperthermia process to be reduced to milliseconds.
We could extract similar conclusions also from some other articles: for example, in ACS Omega 2018, 3, 188−197, they compare the heat generated from bare core Er, Yb:NaGdF4 (5.6 nm) versus the core@shell structure Er, Yb:NaGdF4@NaYbF4 (9.6 nm), in which the later one exhibited two times higher photothermal conversion efficiency while exciting with 975 nm laser, although in that case the spatial separation between the emission/sensing core and the heating shell allowed to tailor the competition between the light and heat generation processes, and enhanced heating capabilities by introducing the rational core/shell design. In another publication (Bioconjugate Chem. 2020, 31, 340-351), NaNdF4 particles of different sizes (4.7 nm, 5.9 nm, 12.8 nm and 15.6 nm) were synthesized via thermolysis reactions. The 12.8 nm sized particles generated more heat, while as smaller the particles are, lower is their ability to generate heat. However, no explanation is given why the 15.6 nm nanoparticles generated less heat than those of 12.8 nm. A recent book chapter (DOI: 10.1007 / 978-3-030-32036-2_12) reaches also the conclusion that bigger particles generate more heat. Nevertheless, this is not always the case. For instance in NaNdF4 particles with sizes of 12 nm, 18 nm and 25 nm (see ACS Appl. Nano Mater. 2020, 3, 2517-2526) less heat is generated as the size of the particles increases. Similarly, from Nanomaterials 2020, 10, 1992, the bare cores Er, Ho, Yb:NaGdF4 (20 nm) generated up to 2 times more heat than the core@shell Er,Ho, Yb:NaGdF4@NaYF4 (23 nm) structures.
Hence, it can be observed from these articles, that a general conclusion about the size or the shape cannot be drawn. We believe that these discrepancies could be related to several factors such as the type of the excitation source (see for example Bioconjugate Chem. 2020, 31, 340-351 in which the particles act different at 793 nm or 808 nm laser source), the power density of the laser source, the excitation with ultrafast pulsed laser sources or with continuous sources, the concentration of nanocrystals in the unit volume and the volume of the solvent taken into account.
Based on these observations, we changed the text in the manuscript as follows, adding also the value of the reported values for the photothermal conversion efficiency of the materials in the references provided above in Table 1 (although we have not been able to incorporate those that only report a temperature increment, since it is not possible to compare with the rest):
“Up to date, there are no reports concerning a size and shape dependence study on the photothermal conversion efficiency of lanthanide doped materials. So, The data in the literature about the effect of the size of different lanthanide doped luminescent nanoparticles as photothermal agents on their photothermal conversion efficiency indicate that a conclusive statement cannot be drawn. [75-78] For example, NaNdF4 particles of different sizes (4.7 nm, 5.9 nm, 12.8 nm and 15.6 nm) exhibit different photothermal conversion efficiency.[75,77] The maximum efficiency was observed for particles with sizes of 12.8 nm, while this efficiency decreased as the size of the particles increased or decreased, indicating that it seems to be an optimal size to maximize the heat generation. Instead, other publications that incorporate a theoretical model, predict that at a given pump power, larger particles of DyPO4 nanoparticles heat the environment more efficiently than the smaller ones when excited with ultrafast pulsed laser sources.[78] Nevertheless, smaller particles have a shorter relaxation time, they heat up faster (albeit to a lower temperature) and cool faster, allowing the inertia of the hyperthermia to be reduced to milliseconds. Such discrepancies can be attributed to several factors that influence the determination of the photothermal conversion efficiency such as the type, wavelength and power density of the excitation source,[77] the concentration of nanocrystals in the unit volume, and the volume of the solvent taken into account.[79]”
References:
- Xu, L.; Li, J.; Lu, K.; Wen, S.; Chen, H.; Shahzad, M.K.; Zhao, E.; Li, H.; Ren, J.; Zhang, J., et al. Sub-10 nm NaNdF4 nanoparticles as near-Infrared photothermal probes with self-temperature feedback. ACS Appl. Nano Mater. 2020, 3, 2517-2526.
- Shao, Q.; Yang, Z.; Zhang, G.; Hu, Y.; Dong, Y.; Jiang, J. Multifunctional lanthanide-doped core/shell nanoparticles: integration of upconversion luminescence, temperature sensing, and photothermal conversion properties. ACS Omega 2018, 3, 188-197.
- Ding, L.; Ren, F.; Liu, Z.; Jiang, Z.; Yun, B.; Sun, Q.; Li, Z. Size-dependent photothermal conversion and photoluminescence of theranostic NaNdF4 nanoparticles under excitation of different-wavelength lasers. Bioconjugate Chem. 2020, 31, 340-351.
- Fedorenko, S.G..; Romanishjun, I.D..; Vanetsev, A.S.; Sildos, I.; Ryabova, A.V..; Loschennov, V.B.; Orlovskii, Y.V. Heating and cooling transients in the DyPO4 nanocrystals under femtosecond laser irradiation in the NIR spectral range. Phys. Wave Phenom. 2018, 3, 198-206.
- Marciniak, L.; Kniec, K.; Elzbieciak-Piecka, K.; Bednarkiewicz, A. In Non-plasmonic NIR-activated photothermal agents for photothermal therapy. Eds. A. Benayas, E.H., G. Hong and D. Jaque, Ed. Springer: 2020; 10.1007/978-3-030-32036-2_12pp. 305-347.
4) Figure: 9. Temperature of 315 °C is insufficient for medical treatment.
Reply: According to the information found in the literature (see for instance Nanoscale, 2014, 6, 9494-9530), the heat that is generated by our Ho, Tm doped KLuW nanoparticles is within the region of the so-called “hyperthermia”, from 41-48 °C (314-321 K). Furthermore, hyperthermia treatments are usually applied in combination with other cancer treatments, such as radiation therapy or chemotherapy, whose efficacy is increased when applied after a hyperthermia cycle (see for example J. Photochem. Photobiol., B, 1999, 53 , 103-109, or Crit. Rev. Bioeng., 2006, 34 , 491-542).
5) The presence of organic surfactants (oleic acid and oleylamine) does not allow the use of NPs for real medical purposes due to their toxicity. Please, comment.
Reply: The final products of the reactions (either microwave-assisted or conventional autoclave) are not coated with organic surfactants, since the calcination process applied at 1023 K removes any organic surfactants present on the surface of the nanoparticles. The only products that are coated with organic surfactants are the seeds, before being treated under calcination at 1023 K, leading to final products which are water soluble, and thus free of organic surfactants (refer to Scheme 1 on the main manuscript). The fact that the final products are not coated with organic surfactants has been verified by the FT-IR spectra of the seeds and the KLuW nanoparticles, as can be seen in Figure S17 in the Supporting Information. This was already indicated in the main text of the manuscript, although we noticed that it was an error in the number of the Figure in the Supporting Information to which the reader was directed. This has been corrected.
6) The article turns out to be too long, but at the same time there is no necessary in-depth analysis of the experimental results.
Reply:
We made an effort to shorten as much as possible the length of the main text of the article, placing all the information concerning the optimization of the synthesis process in the Supporting Information, given that it is the first time that the synthesis of KLuW nanoparticles by solvothermal methods is presented. Nevertheless, the key information to understand the synthesis process had to be kept in the main text of the article. For the characterization of the spectroscopic properties, and the performance of the nanoparticles as luminescent thermometers operating in the third biological window and as photothermal agents, all the information (with all the analysis required to understand the results and be able to compare the performance of these nanoparticles with others reported in the literature) had to be kept in the main text of the article. Thus, we do not agree with the comment of the reviewer, since we made all the efforts possible to shorten the text and the information of the main text to the maximum we could, and we performed all the needed analysis of the experimental results obtained to extract the conclusions provided in the paper.
Reviewer 3 Report
Review for the manuscript:
Entitled: "Effect of the size and shape of Ho, Tm:KLu(WO4)2 nanoparticles in their self-assessed photothermal properties"
for Nanomaterials.
With ID: nanomaterials-1025924
Dear authors,
Thank you for your manuscript.
General comments
Comments for the Authors
This work is well within the scope of Nanomaterials and it may be of interest to most of the readers of this journal. It is well organized with good references to follow. Every section of the manuscript contains enough text to understand the significance of this work and the figures are informative. Some minor issues should be addressed as listed in the specific comments below. For all the above I have opted to recommend a Minor Revision for the current form of the manuscript.
Specific comments
P2, L46: ‘by pumping from outside’. Please explain this and provide appropriate reference.
P4, L187: ‘sample was rotated to gain statistics’ Please explain.
P5, L223: ‘resolution HIGH3’. ‘HIGH3’ refers to resolution or a sensitivity setting?
P5, L231: ‘integrating sphere’ Please provide the model of the integrating sphere.
P5, L231: ‘A glass cuvette containing an aqueous solution…was placed inside the integrating sphere’ Please explain why the glass cuvette was placed inside the sphere and not in the level of the input port? The optical parameters of the cuvette were taken into consideration? Did authors calibrated the sphere to consider the sphere throughput?
P13, L511: ‘With the increase of the temperature, the intensity of the Ho3+ emission band decreases ... due to the thermal activation of luminescence quenching mechanisms,… while the intensity, hence they can be used as a reference for … thermal sensing.’ However, this behavior is common for most scintillating materials.
P22, L772: ‘emissions generated by the nanocrystals, transmitted through the 2 mm thick chicken breast….showing that the biological tissue does not affect the measurements of the temperature by attenuating one of these emission bands more than the others’ This is an important finding. Did authors examine the influence of thicker than 2mm tissues?
Author Response
1) P2, L46: ‘by pumping from outside’. Please explain this and provide appropriate reference.
Reply: Nanoparticles that can generate heat and sense the temperature upon excitation or pumping with laser sources can be embedded in biological tissues for biomedical applications. Thus, these particles that are within the biological tissue can be excited with an external source. See for example, this review on the application of different nanoparticles for photothermal therapy: Nanoscale, 2014, 6, 9494-9530. To avoid misunderstanding, we changed the word “pumping” with “exciting” in the main manuscript, and we added the reference Nanoscale, 2014, 6, 9494-9530, in the end of the corresponding sentence.
2) P4, L187: ‘sample was rotated to gain statistics’ Please explain.
Reply: During the X-ray analysis, rotating our samples will allow the detector to collect more planes than if we simply leave it sitting in one orientation during the scan. Nevertheless, since this a common practice when collecting X-ray powder diffraction data, we removed this expression to avoid misunderstandings.
3) P5, L223: ‘resolution HIGH3’. ‘HIGH3’ refers to resolution or a sensitivity setting?
Reply: HIGH3 refers to the sensitivity setting of the optical spectrum analyzer used. We substituted this expression by the measurement conditions:
“a resolution of 0.5 nm, a level of accuracy of 1 dB, and an integration time of 1 s”
4) P5, L231: ‘integrating sphere’ Please provide the model of the integrating sphere.
Reply: The model of the integrating sphere is the 4P-GPS-010-SL, purchased from Labsphere. We added this information in the main manuscript, together with the characteristics of the integrating sphere:
“A Labsphere 4P-GPS-010-SL integrating sphere with an inner diameter of 1 inch, and 4 ports located at 90º one to each other with a diameter of 0.25 inches was used for this purpose.”
5) P5, L231: ‘A glass cuvette containing an aqueous solution…was placed inside the integrating sphere’ Please explain why the glass cuvette was placed inside the sphere and not in the level of the input port? The optical parameters of the cuvette were taken into consideration? Did authors calibrated the sphere to consider the sphere throughput?
Reply: We used the integrating sphere and the measurement procedure we developed, explained and tested in some other publications. So, here we are reproducing a measurement method that has been already validated, it is nothing new. So, the reviewer and the future reader are kindly asked to refer to the corresponding references. Just for orientation, reference number 44 (Carbon 103 (2016) 134-141) is indicated here. As indicated in this publication all the optical phenomena that might take place when illuminating the sample in the integrating sphere (scattering, reflected, transmittance, absorbance) are taken into account. Nevertheless, when we say inside the sphere, we refer at the level of the input port. To avoid misunderstandings, we modified the corresponding sentence to:
“A glass cuvette containing an aqueous solution of the Ho, Tm doped KLuW nanoparticles with a concentration of 1 g L-1 was placed on the input port of the integrating sphere…”
6) P13, L511: ‘With the increase of the temperature, the intensity of the Ho3+ emission band decreases ... due to the thermal activation of luminescence quenching mechanisms,… while the intensity, hence they can be used as a reference for … thermal sensing.’ However, this behavior is common for most scintillating materials.
Reply: Luminescence thermometry is based in correlating any of the parameters of the luminescence generated by the material with the temperature. In our case, we are using two different emitting centers, and the evolution of the intensity of the emission of these two emission centers is different. Thus, by calculating the intensity ratio between the emissions of these two emission centers, we can generate a robust, self-referred thermometric parameter. Each material, and each pair of emitting centers would provide a different thermal sensitivity and temperature resolution, so then, it is worth to investigate different materials and different pairs of emitting centers to improve the performance of these luminescent thermometers for different applications. This procedure is not new, and the reviewer is kindly asked to refer to any of the excellent reviews and articles published during the last years on luminescence thermometry, some of them highlighted in the references of the manuscript, such as references 12-17 and 23, 33, 35, 36, 41, 44, 66, 69-76, 82, 83, 91-95 .
7) P22, L772: ‘emissions generated by the nanocrystals, transmitted through the 2 mm thick chicken breast….showing that the biological tissue does not affect the measurements of the temperature by attenuating one of these emission bands more than the others’ This is an important finding. Did authors examine the influence of thicker than 2mm tissues?
Reply: We have investigated also the influence of thicker biological tissues, nevertheless we are limited at this thickness because we are applying a limited excitation laser power the goal of not damaging the tissue. The 2 mm thickness allowed us to record a good signal of the photoluminescence arising from the nanoparticles. Nevertheless, in the literature, using other wavelengths, different excitation powers, and different biological tissues, deeper penetration depths could be achieved, up to 1 cm in pork flesh. The reviewer is kinly asked to refer to Savchuk et al., J. Alloys Compnd. 746 (2018) 710-719, where we published a comparison of the penetration depth achieved at that time in different biological tissue samples using different luminescent thermometers.
Round 2
Reviewer 1 Report
The authors addressed my previous comments carefully.
It can be re-considered to be accepted, once authors improve their introduction part with more relevant references and their writing of English.
Author Response
English has been revised all along the manuscript. Concerning the improvement of the introduction part by incorporating more relevant references, we have substituted some of the articles referenced with more relevant reviews. For example, we have eliminated:
Dickerson, E.B.; Dreaden, E.C.; Huang, X.; El-Sayed, I.H.; Chu, H.; Pushpanketh, S.; McDonald, J.F.; El-Sayed, M.A. Gold nanorod assisted near-infrared plasmonic photothermal therapy (PPTT) of squamous cell carcinoma in mice. Cancer Lett. 2008, 269, 57-66.
Shao, J.; Xie, H.; Huang, H.; Li, Z.; Sun, Z.; Xu, Y.; Xiao, Q.; Yu, X. F.; Zhao, Y.; Zhang, H.J. Biodegradable black phosphorus-based nanospheres for in vivo photothermal cancer therapy. Nat. Commun. 2016, 7, 1-13.
Gnach, A.; Lipinski, T.; Bednarkiewicz, A.; Rybka, J.; Capobianco, J.A. Upconverting nanoparticles: assessing the toxicity. Chem. Soc. Rev. 2015, 44, 1561-1584.
Zhang, F. Photon upconversion nanomaterials; Springer: 2015; Vol. 416.
Kolesnikov, I.; Golyeva, E.; Kalinichev, A.; Kurochkin, M.; Lähderanta, E.; Mikhailov, M. Nd3+ single doped YVO4 nanoparticles for sub-tissue heating and thermal sensing in the second biological window. Sens. Actuators B Chem. 2017, 243, 338-345
Savchuk, O.; Carvajal, J.; De la Cruz, L.; Haro-Gonzalez, P.; Aguilo, M.; Diaz, F. Luminescence thermometry and imaging in the second biological window at high penetration depth with Nd:KGd (WO4)2 nanoparticles. J. Mater. Chem. C 2016, 4, 7397-7405
Takei, Y.; Arai, S.; Murata, A.; Takabayashi, M.; Oyama, K.; Ishiwata, S.i.; Takeoka, S.; Suzuki, M. A nanoparticle-based ratiometric and self-calibrated fluorescent thermometer for single living cells. ACS Nano 2014, 8, 198-206
Vetrone, F.; Naccache, R.; Zamarrón, A.; Juarranz de la Fuente, A.; Sanz-Rodríguez, F.; Martinez Maestro, L.; Martín Rodriguez, E.; Jaque, D.; García Solé, J.; Capobianco, J.A. Temperature sensing using fluorescent nanothermometers. ACS Nano 2010, 4, 3254-3258
Martín Rodríguez, E.; López-Peña, G.; Montes, E.; Lifante, G.; García Solé, J.; Jaque, D.; Diaz-Torres, L.A.; Salas, P. Persistent luminescence nanothermometers. Appl. Phys. Lett. 2017, 111, 081901-081907
Zhang, Z.; Suo, H.; Zhao, X.; Guo, C. 808 nm laser triggered self-monitored photo-thermal therapeutic nano-system Y2O3:Nd3+/Yb3+/Er3+@SiO2@Cu2S. Photonics Res. 2020, 8, 32-38.
Ma, L.; Liu, Y.; Liu, L.; Jiang, A.; Mao, F.; Liu, D.; Wang, L.; Zhou, J. Simultaneous activation of short-wave infrared (SWIR) light and paramagnetism by a functionalized shell for high penetration and spatial resolution theranostics. Adv. Funct. Mater. 2018, 28, 1705057-1705068.
And we have incorporated:
Fernandes, N.; Rodrigues, C.F.; Moreira, A.F.; Correia, I.J. Overview of the application of inorganic nanomaterials in cancer photothermal therapy. Biomater. Sci. 2020, 8, 2990-3020.
Khafaji, M.; Zamani, M.; Golizadeh, M.; Bavi, O. Inorganic nanomaterials for chemo/photothermal therapy: a promising horizon on effective cancer treatment. Biophys. Rev. 2019, 11, 335-352.
Cheng, L.; Wang, C.; Feng, L.; Yang, K.; Liu, Z. Functional nanomaterials for phototherapies of cancer. Chem. Rev. 2014, 114, 10869-10939.
Jaque, D.; Martinez Maestro, L.; Del Rosal, B.; Haro-González, P.; Benayas, A.; Plaza, J.; Martín Rodríguez, E.; Solé, J. Nanoparticles for photothermal therapies. Nanoscale 2014, 6, 9494-9530.
Quintanilla, M.; Liz-Marzán, L.M. Guiding rules for selecting a nanothermometer. Nano Today 2018, 19, 126-145.
Naccache, R.; Yu, Q.; Capobianco, J.A. The fluoride host: nucleation, growth, and upconversion of lanthanide-doped nanoparticles. Adv. Opt. Mater. 2015, 3, 482-509.
Reviewer 2 Report
I do not agree with the answer on the following comments
1) Abstract “revealing that small nanocrystals generate more heat and sense the temperature less” This is contrary to previous experiment and theory when larger nanocrystals produce more heat. [S. G. Fedorenko, et. al, Heating and cooling transients in the DyPO4 nanocrystals under femtosecond laser irradiation in the NIR spectral range, Physics of Wave Phenomena, 26, No. 3 (2018) 198–206, ISSN 1541-308X DOI:10.3103/S1541308X18030044]
The correct answer would be: Figure 3 shows a TEM photo of nanoparticles and it can be seen that both form agglomerates. However, small nanoparticles in the volume are located in the form of one-dimensional agglomerates, and large ones in the form of three-dimensional agglomerates. Therefore, despite the fact that in the second method the particles are larger and therefore (with equal pumping) have advantages in the heating temperature (it is proportional to the radius) and in the magnitude of the heat flux into the environment (it is proportional to the volume of the nanoparticle), strong agglomeration in this system does not allow effective use of their more advanced thermophysical properties (large surface and volume) for heating the environment. Whereas small particles in solution in the form of branched filaments are much less agglomerated and have better contact with the environment. This is the reason for their greater heat transfer efficiency. The reason is not in the size of nanoparticles, but in the degree of agglomeration of nanoparticles in solution.
2) Please, give the derivation of Eq. (1) and (2) in Supplement, because the references are unavailable. Now, it is impossible to check the correctness of the idea for temperature measurement and the equations.
To describe the temperature dependence of the multiphonon relaxation rate, the authors use the temperature dependence for a strong electron-phonon interaction, which is not true for the case of a super-weak electron-phonon interaction, which is characteristic of rare-earth ions in optical crystals. See for example, Yu.V.Orlovskii, T.T.Basiev, I.N.Vorob'ev, E.O.Orlovskaya, N.P.Barnes, S.B.Mirov, Temperature dependencies of excited states lifetimes and relaxation rates of 3- 5 phonon (4- 6 mkm) transitions in the YAG, LuAG and YLF crystals doped with holmium, thulium, and erbium, Optical Materials, 18, p. 355- 365 (2002)
Thus, the manuscript requires major revision.
Author Response
I do not agree with the answer on the following comments
- Abstract “revealing that small nanocrystals generate more heat and sense the temperature less” This is contrary to previous experiment and theory when larger nanocrystals produce more heat. [S. G. Fedorenko, et. al, Heating and cooling transients in the DyPO4 nanocrystals under femtosecond laser irradiation in the NIR spectral range, Physics of Wave Phenomena, 26, No. 3 (2018) 198–206, ISSN 1541-308X DOI:10.3103/S1541308X18030044]
The correct answer would be: Figure 3 shows a TEM photo of nanoparticles and it can be seen that both form agglomerates. However, small nanoparticles in the volume are located in the form of one-dimensional agglomerates, and large ones in the form of three-dimensional agglomerates. Therefore, despite the fact that in the second method the particles are larger and therefore (with equal pumping) have advantages in the heating temperature (it is proportional to the radius) and in the magnitude of the heat flux into the environment (it is proportional to the volume of the nanoparticle), strong agglomeration in this system does not allow effective use of their more advanced thermophysical properties (large surface and volume) for heating the environment. Whereas small particles in solution in the form of branched filaments are much less agglomerated and have better contact with the environment. This is the reason for their greater heat transfer efficiency. The reason is not in the size of nanoparticles, but in the degree of agglomeration of nanoparticles in solution.
Reply: We agree with the comment of the referee, and we changed our reasoning according to what he/she is proposing. In addition, we removed the references related to the plasmonic structures, as the mechanism of the generation of the heat is different from that in lanthanide doped materials. Hence, the final discussion in the manuscript read as:
““If we compare the morphology of these nanodimensional Ho, Tm:KLuW particles (see Figure 3), it can be noted that they tend to form agglomerates. However, small nanoparticles in the volume are located in the form of one-dimensional agglomerates, while larger ones are located in the form of three-dimensional agglomerates. Therefore, although the CA methodology produces larger particles, which should generate more heat under equal pumping conditions, as the magnitude of heat flux into the environment is proportional to the volume of the nanoparticles [S. G. Fedorenko, et. al, Heating and cooling transients in the DyPO4 nanocrystals under femtosecond laser irradiation in the NIR spectral range, Physics of Wave Phenomena, 26, No. 3 (2018) 198–206, ISSN 1541-308X DOI:10.3103/S1541308X18030044] and Marciniak from 10.1007 / 978-3-030-32036-2_12), the strong agglomeration observed in this system does not allow an effective use of their more advanced thermo-physical properties (large surface and volume) for heating the environment. On the contrary, small particles in solution in the form of branched filaments are much less agglomerated, therefore have better contact with the environment, which leads to higher heat transfer efficiency. This conclusion can be confirmed furthermore if we compare also the results obtained from the particles synthesized by the modified sol-gel Pechini method (see Figure 7). These particles tend to form big agglomerates, bigger than those of the nanoparticles synthesized via the solvothermal methodologies, leading to smaller heat efficiencies.”
- Please, give the derivation of Eq. (1) and (2) in Supplement, because the references are unavailable. Now, it is impossible to check the correctness of the idea for temperature measurement and the equations.
To describe the temperature dependence of the multiphonon relaxation rate, the authors use the temperature dependence for a strong electron-phonon interaction, which is not true for the case of a super-weak electron-phonon interaction, which is characteristic of rare-earth ions in optical crystals. See for example, Yu.V.Orlovskii, T.T.Basiev, I.N.Vorob'ev, E.O.Orlovskaya, N.P.Barnes, S.B.Mirov, Temperature dependencies of excited states lifetimes and relaxation rates of 3-5 phonon (4- 6 mkm) transitions in the YAG, LuAG and YLF crystals doped with holmium, thulium, and erbium, Optical Materials, 18, p. 355- 365 (2002)
Thus, the manuscript requires major revision.
Reply: The objective of the present manuscript was not to develop a new model for the determination of the thermometric performance of these nanothermometers, but apply one that was previously developed and has not been questioned up to now. The model we used in this manuscript has been used by other authors in the literature without objection (see for instance Pandey, A.; Rai, V.K., Optical thermometry using FIR of two close lying levels of different ions in Y2O3:Ho3+-Tm3+-Yb3+ phosphor, Appl. Phys. B, 113 (2013) 221-225; Brites, C.D.S.; Lima, P.P.; Silva, N.J.O.; Millan, A.; Amaral, V.S.; Palacio, F.; Carlos, L.D., Thermometry at the nanoscale using lanthanide-containing organic-inorganic hybrid materials, J. Lumin., 133 (2013) 230-232), even by very well known theoretical developers of lanthanide spectroscopy, such as Prof. Andries Meijerink (see for instance his seminar about this subject at http://nanotbtech.web.ua.pt/videos/video1.mp4). Furthermore, this model allow us to compare the performance of the thermometers we developed with those previously reported in the literature. This model has been developed for two emitting Ln3+ ions indistinguishably employed as the probe or the reference, in which ion-ion energy transfer regulated the changes in the intensity ratio. In the model, the authors followed the discussion of Cooke et al. (Cooke, D.W.; Muenchausen, R.E.; Bennet, B.L.; McClellan, K.J.; Portis, A.M., Temperature-dependent luminescence of cerium-doped ytterbium oxyorthosilicate, J. Lumin. 79 (1998) 185-190; Cooke, D.W.; Bennet, B.L.; Muenchausen, R.E.; Lee, J.K.; Nastasi, M.A., Intrinsic ultraviolet luminescence from Lu2O3, Lu2SiO5 and Lu2SiO5:Ce3+, J. Lumin., 196 (2004) 125-132) that extends the classical Mott-Seitz model for competition between radiative and nonradiative transitions within a luminescence center, and taking into account the particular case in which the exponential terms dominate the intensity of each transmission, i.e. 1<< αjexp (-ΔEj / kB T) with j = 1, 2, in which α is the ratio between the nonradiative and the radiative transition probabilities, ΔE is the activation energy of the thermal quenching process, kB is the Boltzmann constant and T is the temperature. The most remarkable feature of this approximation is that the resulting equation has the same functional form than that derived from the Boltzmann law for single emitting center, and thus, although we are dealing with a dual-center emission the problem is reduced to the single-center case.
Thus, if the reviewer considers that this model is not correct, it should address a comment to the original publication where this model was firstly presented (Brites, C.D.S.; Millán, A.; Carlos, L.D. Chapter 281-Lanthanides in Luminescent Thermometry. In Handbook on the Physics and Chemistry of Rare Earths, Jean-Claude, B., Vitalij K, P., Eds. Elsevier: 2016; Vol. 49, pp. 339-427).